# Utilization of Legume-Nodule Bacterial Symbiosis in Phytoremediation of Heavy Metal-Contaminated Soils

**DOI:** 10.3390/biology11050676

**Published:** 2022-04-27

**Authors:** Monika Elżbieta Jach, Ewa Sajnaga, Maria Ziaja

**Affiliations:** 1Department of Molecular Biology, The John Paul II Catholic University of Lublin, Konstantynów Street 1I, 20-708 Lublin, Poland; 2Laboratory of Biocontrol, Application and Production of EPN, Centre for Interdisciplinary Research, The John Paul II Catholic University of Lublin, Konstantynów Street 1J, 20-708 Lublin, Poland; esajnaga@kul.pl; 3Institute of Physical Culture Studies, Medical College, University of Rzeszów, Cicha Street 2a, 35-326 Rzeszów, Poland; mziaja@ur.edu.pl

**Keywords:** biorhizoremediation, heavy metals, legumes, phytoextraction, phytoremediation, phytostabilization, plant growth-promoting, rhizobia

## Abstract

**Simple Summary:**

The legume–rhizobium symbiosis is one of the most beneficial interactions with high importance in agriculture, as it delivers nitrogen to plants and soil, thereby enhancing plant growth. Currently, this symbiosis is increasingly being exploited in phytoremediation of metal contaminated soil to improve soil fertility and simultaneously metal extraction or stabilization. Rhizobia increase phytoremediation directly by nitrogen fixation, protection of plants from pathogens, and production of plant growth-promoting factors and phytohormones.

**Abstract:**

With the increasing industrial activity of the growing human population, the accumulation of various contaminants in soil, including heavy metals, has increased rapidly. Heavy metals as non-biodegradable elements persist in the soil environment and may pollute crop plants, further accumulating in the human body causing serious conditions. Hence, phytoremediation of land contamination as an environmental restoration technology is desirable for both human health and broad-sense ecology. Legumes (*Fabaceae*), which play a special role in nitrogen cycling, are dominant plants in contaminated areas. Therefore, the use of legumes and associated nitrogen-fixing rhizobia to reduce the concentrations or toxic effects of contaminants in the soil is environmentally friendly and becomes a promising strategy for phytoremediation and phytostabilization. Rhizobia, which have such plant growth-promoting (PGP) features as phosphorus solubilization, phytohormone synthesis, siderophore release, production of beneficial compounds for plants, and most of all nitrogen fixation, may promote legume growth while diminishing metal toxicity. The aim of the present review is to provide a comprehensive description of the main effects of metal contaminants in nitrogen-fixing leguminous plants and the benefits of using the legume–rhizobium symbiosis with both wild-type and genetically modified plants and bacteria to enhance an efficient recovery of contaminated lands.

## 1. Introduction

Legumes (family *Fabaceae*), i.e., species that produce seeds in a pod, constitute the third largest family of flowering plants, consisting of more than 20,000 species. They include herbaceous plants, trees, shrubs, and creepers, with only a limited number used as human food [1]. Meeting the increasing demand for food is now an increasingly prominent issue. In this context, legume crops can play a significant role in human and animal nutrition [2]. The most common legumes used for human consumption include the following: faba bean (*Vicia faba* L.); bean (*Faseolus vulgaris* L.); mung bean (*Vigna radiata* L.), peas (*Pisum sativum* L.); chickpea (*Cicer arietinum* L.); lentis (*Lens esculenta* Medik.); peanuts (*Arachis hypogaea* L.); lupin (*Lupinus* ssp.); cowpea (*Vigna anguiculata* L.); soybean (*Glycine max* L.) [3,4]. They provide a sustainable and affordable alternative to meat and are considered the second most important food source after cereals. They are an excellent source of superior quality protein with 20–45% of protein, complex carbohydrates (±60%) and dietary fiber (5–37%). Legumes do not have cholesterol and are generally low in fat, except for soybeans (±47%), peanuts (±45%) and chickpeas (±15%) and provide essential minerals and vitamins [5]. Furthermore, legumes also have medicinal functions due to their content of beneficial bioactive compounds, which play a role in the prevention of some diseases [6,7]. Increased consumption of grain legumes, and dietary supplements based on legumes, is associated with numerous beneficial health effects [8,9,10]. 

Legumes are also well known to offer indirect/direct benefits to agroecosystems. Numerous studies confirm that legumes can be cover crops (CCs). Legume CCs exert several positive effects, i.e., they improve soil fertility, prevent soil erosion, provide soil protection, and reduce weed infestation [11,12,13]. Legume crops are commonly used as green manure [14]. They enrich the soil with nitrogen (N) and thus provide a better environment for subsequent crops improving their growth and productivity [15]. Legumes can fix substantial amounts of free atmospheric nitrogen via symbiosis with nitrogen-fixing rhizobacteria (NFB) [16]. The availability of nitrogen from NFB reduces the need for N fertilizers in the legume crop to almost zero, saving costs related to the purchase and application of fertilizers (mineral or organic) [7,17]. The inclusion of legumes in crop rotations is a sustainable approach to reducing N fertilizer requirements and increasing subsequent crop yields [18,19]. Legume CCs reduce erosion risks by physically protecting the soil surface, improving soil structural properties, and increasing the soil organic carbon (C) concentration. An increase in organic C in the soil is one of the main factors contributing to increased soil stability, because organic C can bind soil particles and form stable macroaggregates [20]. In addition, legumes can be a useful means to control weeds in agroecosystems [11]. They can be planted with the crop or planted during the fallowing period, providing weed suppression through competition for light, soil water, and nutrients by creating a physical and chemical barrier [21,22]. Due to their environmental and socioeconomic benefits, legumes can be introduced in modern cropping systems to increase crop diversity and reduce use of external inputs [3,23]. 

Among the many environmental benefits provided by legumes, climate mitigation is worth mentioning, as these plants can contribute to reduction in the emission of greenhouse gases (GHG), such as carbon dioxide (CO_2_) and nitrous oxide (N_2_O), compared to agricultural systems based on mineral N fertilization [24]. Legume crops emit around five to seven times less GHG per unit area compared to other crops, and significantly reduce N_2_O emission [25]. Legumes are considered appropriate crops for the recovery of marginal lands, especially heavy metal contaminated soils, through symbiosis with *Rhizobacteria*. They can be used as pioneer plants in areas where other plants are unable to grow [20,26]. 

The contamination of soil with heavy metals (HMs) is one of the most pressing environmental problems in Europe and throughout the world, especially in highly industrialized regions. HMs are non-degradable and can persist in the soil for a prolonged period, which poses a long-term threat for the environment. The industrialized economy, human population growth, and anthropogenic activities have led to a significant increase in the production and release of HMs into the environment in recent decades [27,28,29,30]. The nature of pollution is quite varied north-east Europe and eastern-central Europe are less affected by high concentrations of HMs, while most soil samples in Western Europe and the Mediterranean have concentrations above the threshold values of at least one kind of HM [31]. Contamination of soil with HMs poses a significant risk to both agroecosystems and human health through the food chain [32]. Phytoremediation, i.e., the use of plants to clean up contaminated areas, has emerged as an alternative to address this global concern [33,34]. Several legumes, especially *Anthyllis, Cytisus, Lotus, Lupinus, Genista, Glycine, Ononis, Ornithopus, Pisum*, several species of *Trifolium,* and *Vicia*, have been proposed as relevant species for phytoremediation [35], which in this case is called biorhizoremediation [36]. This technology is a natural and environmentally friendly method offering a real economic alternative for HM removal in contrast to physicochemical processes [37,38]. HMs in the environment are of both natural and anthropogenic origin. Natural sources of heavy metal contamination include rock weathering, soil erosion, forest fires, and volcanic activity; however, the most important sources are anthropogenic processes, including extensive mining, smelting, metallurgy, the industrial wastewater discharge, and coal combustion. Agriculture is an additional source of HM pollution, especially the use of chemical fertilizers and pesticides (herbicides, insecticides, fungicides) [37,39,40,41,42]. Unlike organic pollutants, HMs once introduced into the environment cannot be broken down or biodegraded. Therefore, increased bioaccumulation of HMs is now being perceived as an imminent threat to the ecosystem and environment [43,44,45]. Numerous studies show the negative effect of HMs on the biological properties of soil, which in turn affect its long-term chemical and physical properties and the ability to support plant growth [46,47,48]. HMs affect the diversity and activity of soil microorganisms, inhibiting their growth and reproduction rate [49,50]. Moreover, the activity of enzymes in sites contaminated with HMs is reduced [51,52]. Contamination of soil by HMs is a global issue of growing importance due to their toxicity to human health [53,54,55,56]. Metals accumulate in the food chain through uptake at the primary producer level (plant roots take up HM ions) and then through consumption at the consumer level [57]. 

Ingestion of plant contaminated with HMs causes serious human health problems, such as carcinogenic effects, weak immunological mechanisms, mental growth retardation, cardiovascular complications, and other serious diseases of the liver, lung, kidney, and nervous system [58,59]. They additionally have an impact on cell organelles and components, such as cell membrane, mitochondria, nuclei, and lysosomes. Metal ions have emerged to interact with DNA and nuclear proteins, causing DNA damage [60]. The key roles in toxicity and carcinogenicity are played by arsenic (As), cadmium (Ca), chromium (Cr), lead (Pb), and mercury (Hg). Due to their high degree of toxicity, they belong to the priority metals for human health and cause multiple organ damage, even at lower levels of exposure [61]. 

Attention is increasingly being paid to methods of HM removal from soil that are beneficial to the environment and contribute to sustainable development. These include the use of bioremediation carried out by microorganisms, plants, or a combination of both organisms for the treatment of HM polluted soils [62,63]. The emphasis is that phytoremediation-based strategies are green technologies that are cost-effective and can be used as long-term solutions [64,65,66]. Thus, the aim of this review is to provide comprehensive information about the main effects of metal contaminants on nitrogen-fixing leguminous plants, the metal tolerance/resistance mechanism of the legume–rhizobium symbiotic system, including the use of genetically modified symbionts, in HM contaminated soils, and the benefits of biorhizoremediation in efficient restoration of HM contaminated lands.

## 2. Toxic Effect of Metal Stress on Plants

Heavy metals in low concentration play an important role in plant development processes. Some of them for example cobalt (Co), iron (Fe), manganese (Mn), molybdenum (Mo), nickel (Ni), zinc (Zn), and cooper (Cu) are essential micronutrients for proper growth and are involved in important metabolic processes in plants [67,68]. Furthermore, HMs are also considered cofactors of various cellular enzymes, and actively participate in several oxidation-reduction reactions [69,70]. Although HMs are naturally present in soils, when they exceed their threshold concentrations, their actions are considered toxic to plant development [62,71], as presented in Table 1. Some of the direct toxic effects caused by high metal concentrations include the generation of reactive oxygen species (ROS), causing oxidative stress, inhibiting cytoplasmic enzymes, and causing damage to cell structures (e.g., DNA, proteins, lipids) [72,73]. Furthermore, excessive concentrations of HMs inhibit such physiological processes as photosynthesis, respiration, transpiration, mineral nutrition, biomass production and may consequently cause plant death [70,74,75,76]. The toxicity of metals has an adverse effect on the roots, reducing the plant’s ability to absorb water and nutrients. Consequently, disturbance in the functioning of roots and leaves affect such processes as flowering, fruiting, and seed formation [77].

HM ions present in the soil are taken along with nutrients in the water and incorporated into plant tissues. Plant roots play the most vital role in the uptake and translocation of HMs. Metals first enter the root cells with the help of transporters and are then transported into the shoot system [110]. Exposed to HMs, plants have developed mechanisms to detoxify the adverse effect of HMs, i.e., avoidance and tolerance. The avoidance strategy refers to the ability of plants to limit the uptake of HMs. In this strategy, an important role is played by mycorrhizas, extracellular exudates (organic acids and amino acids) and the cell wall. In turn, the tolerance strategy involves intracellular defense, once HM ions enter the cytosol. This is achieved mainly through chelation by complexation of HM ions with ligands. Phytochelatins (PCs) and metallothioneins (MTs) are among the most common peptide ligands [111,112,113,114,115]. In plant–microbe interaction, PCs and MTs as metal-binding proteins (MBPs) increase the accumulation and tolerance/resistance of HMs [116]. During HM binding, the HM absorption occurs. Then, HMs are translocated into roots and shoot from the rhizosphere zone, where their various intracellular mechanisms and cytosolic approaches are used for HM detoxification by PCs [117]. After chelation, the complexes of ligands with HMs are transported to the vacuole where those complexes are stored without toxicity [114,115,116]. In plants, there are four MT genes present: in roots (MT1 gene), in leaves (*MT2* gene), in ripe fruit (*MT3* gene), and in seeds (*PEC/pecMT* gene). Activation of the *MT4* gene causes an increase in tolerance to metal ions, both Cu and Zn, in the leaves and roots of host plant [118,119]. Moreover, during various stress conditions including HM stress, plants and their plant growth-promoting (PGP) rhizobacteria (PGPB) adopt to different mechanisms of defense, such as the compartmentalization, the formation of complexes, exclusion, as well as the secretion of MBPs [116]. 

Plants uptake HMs from the rhizospheric system into roots mainly through two pathways, (1) the symplastic pathway (active transport through the plasma membrane of cell roots via specific ion channels or (2) the apoplastic pathway (passive diffusion) between cell walls [120,121]. The uptake and translocation of HMs in plants take place through specialized transporters located in the plasma membrane of root cells. These transporters represent several families, i.e., zinc–iron permease (ZIP), HM transport ATPase, natural resistant associated macrophage protein (NRAMP), cation diffusion facilitator (CDF) and ATP-binding cassette (ABC) [122,123,124]. Transporters of the ZIP family are involved in HM accumulation processes including the uptake, transport, and detoxification of many cations (e.g., Zn, Fe, Cd, Cu, Mn) from root to shoot [125,126]. ATPases are involved in the transport of HMs in long-distance root-to-shoot translocation (such as Cu, Zn, Cd, Pb) and play a key role in metal homeostasis and tolerance [127]. NRAMP are also involved in the transport of divalent metal ions including Mn, Zn, Cu, Fe, Cd, Ni, and Co, which participate in the control of homeostasis in plant [128,129]. The CDF family is also known as metal tolerance proteins (MTP) involved in the translocation of metals (such as Zn, Cd, Co, Fe, Ni and Mn), from the cytoplasm to the vacuole [130,131]. The ABC protein family serves as membrane-intrinsic active pumps that consume ATP to function. They are involved in transmembrane transport of a wide range of molecules, such as lipids, phytohormones, carboxylates, chlorophyll catabolites, and HMs (As, Cd, Hg, Al) [132]. 

The use of legume plants in phytoremediation of metal-contaminated soils represents a very important area of research exploiting the maximum of *Rhizobacteria* symbiosis advantages [38]. Regarding HMs sensitivity, plants can be classified as tolerant and/or hyperaccumulator species [133,134]. Plants suitable for phytoremediation have important characteristics, such as rapid growth, high biomass, and accumulation of high levels of HMs [135]. The use of some legume plants in biorhizoremediation has been well documented, as presented in Table 2.

Further legume plants are being sought for the use in phytoremediation. *Albizia lebbeck* L., *Bauhinia purpurea* L., *Dalbergia sissoo* Roxb. ex DC., *Milettia peguensis* Ali. and *Pongamia pinnata* may be successfully used for HM phytoextraction processes, especially when they grow in urban forests watered by untreated industrial wastewater containing HMs. These selected tree species showed favorable uptake of HMs, translocation, and HM storage in different plant parts as accumulators, without the addition of chelating agents and/or organic adjustment. Thus, it may be an eco-friendly and sustainable way for reduction in multiple urban problems such as toxicity of untreated industrial wastewater, air pollution through urban forestry [163], and saving fresh water used for irrigation.

## 3. Legume–Rhizobium Symbiotic System for Stress Condition Tolerance

### 3.1. Plant Growth-Promoting Rhizobia Assisted Legumes for Metal Phytoremediation

HM contamination in combination with phosphorus (P) and nitrogen limitation are primary factors inhibiting revegetation of contaminated lands, limiting water retention, and resulting in periodic erosion, and loss of nutrients and organic matter [164,165]. However, some soil features, such as pH, may have a larger influence on individual plant species than HM contamination due to their effect on metal bioavailability [165]. Moreover, HM contamination of field soils is less phytotoxic to plants and symbiotic bacteria than laboratory contaminated and polluted soil at equivalent concentration of total HMs, especially in the case of soils with neutral or alkaline pH (e.g., calcareous). The bacteria are more tolerant in even highly contaminated field soils, and they can be selected for survival in the soil with less bioavailable metals [166,167,168,169]. Hence, legumes with NFB become a natural choice when searching for plants that can grow on devastated and/or contaminated soils with poor nutritional values. The fixation of atmospheric N_2_ into ammonia by NFB as PGPB are essential for plant growth and productivity, especially in soils with N and P deficiency [170]. The genera *Rhizobium* is one of the most effective P-solubilizers [171]. Furthermore, other NFB properties, e.g., the siderophore production to increase Fe uptake, enhance bioavailability of metals and its subsequent removal form soil [172,173], and protection of plants against pathogens [172] as well as release of volatile substances such as acetoin and 2,3-butanediol [174], are important for PGP, especially in a HM contaminated environment, become the basis of using the legume–rhizobium symbiotic system for phytoremediation. In addition, to protect plants and microorganisms from HM phytotoxicity, field soils should be alkaline calcareous [175,176,177]. Interestingly, such NFB as bradyrhizobia alkalize the growth environment [178,179]. In addition, the old, degenerated nodules can be the main nutrient source for both microorganisms and legumes in poor and devastated soils such as calamite tailings [133,137]. 

### 3.2. Nitrogen Fixation Bacteria

NFB are the most numerous representation of the phyto-microbial population in legume nodules, belonging to α-rhizobia (*Alphaproteobacteria* represented by *Bradyrhizobium* and *Rhizobium* genera) and β-rhizobia (*Betaproteobacteria* such as *Burkholderia* and *Cupriavidus*) [180]. Although both rhizobium groups are evolutionarily distant, their symbiotic (*nod* and *nif*) genes are remarkably similar pointing to lateral transfer [181,182,183]. Interestingly, legume root nodules may also contain many different microorganisms (bacteria and fungi called non-rhizobium endophytes, NRE) as an additive to NFB [170,184,185]. Legumes are capable of establishment of mutually beneficial symbioses with either rhizobia or mycorrhiza that provide the host plant with water and minerals. Rhizobial and mycorrhizal invasion share a common plant-specific genetic program controlling the early host interaction [186]. Moreover, fungi may change the bacterial community and function as PGP improving shoot nitrogen, phosphorus and potassium (K) levels [187]. Additionally, N_2_-fixing *Burkholderia* are able to degrade various wastes such as acetone, ethanol, hexane, methylterbutylester, petrol and toluene [188], thus helping to clean up the soil.

The N-fixing effectiveness depends on the NFB strain. Both the nodulation process and the N fixation function are genetically controlled [188]. Many NFB strains can nodulate a wide range of different plant species, while some strains may nodulate even one specific legume plant [179,189,190,191,192,193,194]. The nodulation and N-fixation functions of NFB can be improved by selection of rhizobia and host plants or by genetic modification. In particular, NFB can be selected for their efficiency, competitiveness among natural bacterial populations, and adaptivity to the contaminated environment where they are introduced for nodulation [188]. Noteworthy, lateral gene transfers play a significant role in bacterial adaptation to the living environment. In soils with no legumes, there are non-symbiotic rhizobial strains that can often become effective symbionts upon acquisition of symbiotic genes from inoculant strains [195]. For example, bradyrhizobia are abundant in many field soils where leguminous plants are absent, frequently acting as endophytes of various plants [196]. The symbiotic genes of NFB are conserved. The *nod* and *nif* genes evolved independently in comparison with housekeeping genes, suggesting a different origin. Possibly, they were acquired via lateral transfer. The nodulation *nod* A, B, C genes likely originate from outside, since their G+C concentration is significantly lower than the average G + C concentration of rhizobia [197]. 

### 3.3. Mechanism of the Legume–Rhizobium Interaction

When legumes are planted in field soil, an interaction between the legumes and NFB present in the ground establishes through a complex signal factor exchange initiated by secreting aromatic plant flavonoids. In the soil, the flavonoids are recognized by compatible rhizobial species [198,199,200], which produce lipochitooligosaccharide compounds, also called Nod factors (*myc/nod* factors) which are detected by LysM type receptors on the host legume plant [201,202]. It is worth emphasizing that the rhizobial symbiosis only occurs with legumes and non-legume *Parasponia* plants just through Nod factor recognition by the LysM type receptors [203]. *Bradyrhizobium* is responsible for symbiosis with *Parasponia* plants which suggest that bradyrhizobia may have been the ancestor of all rhizobial bacteria [204,205].

Nod factors and symbiotic exopolysaccharides (EPSs) activate multiple responses in the host leguminous plant preparing the legume to receive the invading rhizobia [206]. This triggers calcium spikes in the inner cortical cells, causing the formation of nodule primordia in dividing root cells [207,208]. The recognition of the Nod factor by the legume plant also causes curling of root hair forming infection threads, which trap the attached rhizobia and create a curled colonized root hair [209]. Legume root cells in the inner cortex incorporate the invading microorganisms in host membrane-bound compartments, which mature into structures called symbiosomes. In the nodule, incorporated bacteria develop into bacteroids that may terminally differentiate into NFB bacteroid forms that are capable of N fixation [198,206]. However, in the nodule there are also undifferentiated bacteroids which are not capable of N fixation. Interestingly, during invasion and symbiosis, rhizobia can avoid innate immune response of the legume plant to survive within the symbiosome compartment [206]. However, the environment inside nodules can be a challenge for NFB due to the alterations in osmolarity, oxygen concentration and oxidative stress, the reduced pH, the influence of plant peptides [198] and the HM intake by plant roots.

### 3.4. The Rhizobial Tolerance to Various Stress Conditions

Bacterial genes involved in adaptation to varying stress conditions inside nodules are critical for the establishment of a functional symbiotic relationship [198]. The presence of high osmotic or salt stress conditions induces transcription of genes involved in nodulation *(nod* and *nif* genes) and N fixation (*fix* genes) by NodD2 in *Rhizobium tropici* strains [210,211]. Additionally, increased salt contents regulate EPS production in *Sinorhizobium meliloti* [212]. Moreover, NFB can produce periplasmatic cyclic β(1-2)-linked glucan when they grow in hypotonic conditions [198]. The inability to produce the glucan through mutating the *ndvB* gene in *Rhizobium meliloti* resulted in sensitivity to hypotonic conditions and inhibited nodule formation in *Medicago truncatula* Gaertn. [213]. However, the production the symbiotically important polysaccharide succinoglycan (EPS-1) by NFB might provide enough osmoprotectant to facilitate survival in the absence of cyclic β(1-2) glucan in *ndvB* mutants [214]. Another way of protection against high osmolarity is the accumulation of ions such as K, by NFB [215,216]. Increased K levels enhance nitrogenase activity in *Bradyrhizobium* sp. (32H1 strain) cultured under low oxygen stress [217]. Therefore, the osmotic stress tolerance is linked with signal regulation of nitrogenase in bacteroids via the regulation of the K content [198,215]. 

Another important bacterial challenge is the low oxygen concentration in the nodule. The strict adjustment of oxygen levels to optimal for N fixation in bacteroids is created by a diffusion barrier [218]. The oxygen regulation leads to several signaling and physiological alterations in the bacterium cell, promoting symbiosis and N fixation. A low oxygen level activates the two-component system (hFixL-FxkR) with 3 proteins (hFix1, FnrN and NifA) acting as oxygen sensors, which enhances the transcription of most genes involved in N fixation [219,220,221]. The hFix1 protein induces the expression of FnrN in the meristem zone (I) and the invasion zone (II), which then induces the expression of *fixNOQP* genes in the N fixation zone (III) when oxygen levels are almost anaerobic creating microaerobic stress conditions [222]. The gene induction in microaerobic conditions is necessary for nitrogenase production, and protein function. Microaerobic stress acts as a signal for legume–rhizobium symbiosis [198]. 

Moreover, in nodules, NFB struggle with ROS, which is another strategy to protect plants. However, ROS generation by legumes is beneficial for establishment of symbiosis [223,224]. ROS are the plant response to Nod factor recognition and are mainly associated with the NADPH oxidase activity [225]. NFB use many mechanisms to eliminate potential damage from ROS, including catalase production [223,226]. However, when catalase is over-expressed in *S. meliloti*, abnormal formation of infection thread and delayed development of nodules are observed [226]. Therefore, ROS may play significant role in the development of infection threads, or they induce physiological alterations in NFB that are necessary for symbiotic process [227]. ROS encoded for *PvRbohB* genes in *Phaseolus vulgaris* are important for symbiosis, especially when NFB exposed to oxidative stress produce increasing quantities of EPS [228,229]. EPS production seems to be involved in the tolerance of different stresses encountered by microorganisms. Moreover, the EPS-1 fraction is responsible for removal of H_2_O_2_ from the environment [230]. Therefore, oxidative stress promotes EPS production, which is necessary for ROS tolerance and essential for symbiotic interaction. The production of H_2_O_2_ by plants is a response to symbiotic formation promoting creation of a symbiotic signal [198].

The ability to tolerate acidic pH stress conditions is particularly important for the use of legumes for biorhizoremediation. During growth, plants can excrete acidic compounds into surrounding field soil, thereby decreasing the pH of the environment by as much as 2 pH units [231]. It is predicted that the bacteroid and peri-bacteroid zones are acidic, reaching a pH of 4.5 [232]. Furthermore, the curled colonized root hair is a space with local acidic stress [233]. The response of NFB to acidic pH is regulated via large networks of multiple genes [234]. This response is predominantly regulated through a two-component RSI system (*actR/S* and *ChvI/exoS/exoR*) which finally controls the regulation of cytoplasmic pH or the production and modification of extracellular compounds for pH tolerance components [235,236,237]. The regulation of intracellular K efflux proteins is significant for pH tolerance. K levels regulating nitrogenase activity and leading to the accumulation of K^+^ in acidic stress function as a symbiotic signal [217]. Additionally, glutathione (GSH), which is involved in tolerance of various stress conditions such as pH and ROS stresses, is produced in high quantities in acidic conditions [238,239]. Furthermore, acidic tolerant NFB strains produce more symbiotically important EPS-1 than acid-sensitive strains in non-stress conditions [240,241]. EPS-1 production regulated by the RSI system is required not only for acidic pH tolerance but also for survival in the nodules and is necessary for symbiotic signaling [198,241,242,243] The succinylation of EPS is also a crucial step for the symbiotic signaling and formation [244]. 

Plants also produce anti-microbial peptides (AMPs) also called nodule-specific cysteine (NCR) peptides with anti-microbial activity against microorganisms including rhizobia [245,246,247]. However, protein BacA, i.e., a transporter for AMPs, in *S. meliloti* and BclA in *Bradyrhizobium* sp. are involved in tolerating the challenge with NCRs. Mutants with *bacA* gene deletion were hypersensitive to NCRs present in nodules [246,248]. NCR recognition may be a signal for NFB facilitating adaptation to conditions inside plants, enhancing the production of polysaccharides necessary for symbiotic interaction and induction of physiological and morphological changes in NFB for differentiation of bacteroids necessary for nitrogen fixation [198]. 

### 3.5. The Legume–Rhizobium Symbiosis in Heavy Metal Stress Response

Using multiple mechanisms, NFB help legumes growing in metal stress conditions through increased HM tolerance/resistance and PGP ability. Rhizobia isolated from native legumes growing in HM contaminated field soils are predominantly HM tolerant. Isolates effective in N fixation are identified as *Rhizobium, Bradyrhizobium*, *Mesorhizobium* and *Sinorhizobium* species tolerant to several HMs and metalloids such as Ni, Co, and Cr [36,149,165,169]. Some strains directly promoted root growth in legumes, e.g., *Lotus corniculatus* L., and non-legume *Arabidopsis thaliana* L. under metal stress. Interestingly, the metal treated nodules showed structural changes, including increased phenol accumulation and wall thickening with higher concentrations of celluloses, calloses, glycoproteins, hemicelluloses, and pectins [169]. NFB are also beneficial to legumes through the synthesis of phytohormones such as abscisic acid (ABA) and indole-3-acetic acid (IAA) [150]. Almost 80% of NFB are capable of producing IAA, an important auxin affecting division and differentiation of plant cells, which stimulates plant growth and performance of nodules in the symbiotic relationship [249]. For example, *Agrobacterium tumefaciens* (CCNWGS0286 strain) isolated from nodules of *Robinia pseudoacacia* L. growing in Zn-Pb mine tailings, was able to overproduce IAA in the presence of Cu and Zn significantly enhancing legume growth in comparison with inoculation of an *A. tumefaciens* mutant with lower IAA production [250]. Moreover, Bianco and Defez [251] showed that recombinant *Rhizobium* species such as *S. meliloti* overproducing IAA, improved the N fixation ability in *Medicago* plants compared to wild-type (WT) strain.

Three IAA synthetic pathways in NFB have been detected: indole-3-acetamide (IAM), tryptamine (Tra), and indole-3-pyruvic acid (IPyA) pathways [252]. IAA induces the activity of some critical enzymes such as citrate synthase and α-ketoglutarate dehydrogenase in the tricarboxylic acid (TCA) cycle which delivers energy necessary for N fixation. *R. leguminosarum* bv. *viciae*, *R. trifolii* and *S. meliloti* mutants deficient in the *dct* gene encoding the dicarboxylic acid transport factor and *S. meliloli* mutants without such genes as *gltA* encoding citrate synthetase and *icd* coding for isocitrate dehydrogenase were unable to fix N_2_ [165]. However, such stress environmental factors as oxidation and dilution can reduce the influence of IAA to the threshold level required for growth promotion [253]. Nonetheless, low IAA concentrations (around 10^−8^ M) exert a beneficial stimulating effect on the improvement of root development as well as nutrient and mineral uptake by plants and enhancement of bacterial colonization and nodulation [254,255]. In turn, a high IAA concentration inhibits nodulation due to the increase in the ethylene level, which halts the growth of the root system [255,256]. Furthermore, plants with nodules contained higher IAA levels than non-nodulated ones [257], and this is partly related to the high production of IAA by NFB in nodules [255]. Therefore, IAA may be involved in controlling the number of nodules [258] and the efficiency of nodulation [259]. It is worth emphasizing that IAA transport may also be affected by Nod factors, and in some NFB such as *R. leguminosarum* and *S. meliloti*, flavonoids were able to promote IAA synthesis, showing that bacterial IAA production and transport could be regulated by specific host legumes [165,252,260]. 

NFB are likely to contain various metal tolerance/resistance systems to maintain metal homeostasis both in the bacteria and in plant cells. For example, some metal resistant species of *Rhizobium* increase the concentrations of some substances such as amino acid-like thiol, EPS, GSH, proline, and urease as well as cell inclusions upon exposure to Cd, Cu, Ni and Zn [261,262,263]. Furthermore, an increase in metal concentrations induces the transfer of naturally occurring plasmids with MH tolerance/resistance genes between NFB. The plasmid transfer plays a key role in conferring augmented metal tolerance/resistance [264]. However, there are many genes related to metals resistance/homeostasis in whole genome sequences [165]. NFB isolated from natural sites contaminated with HMs often present tolerance to multiple pollutants as they have adapted to stressful conditions [265]. This adaptation is most likely related to the acquisition of genes responsible for resistance to HMs and stimulation of broad-legume-range resistance plasmid transfer to bacteria in HM contaminated site and/or promoting the efficient transformation and incorporation of crucial genes for the adaptation [266]. In bacteria, the resistance genes such as *cnr/nre*, *PbrA* and *CadA, cadA* (PIB-2-ATPase), *nreB* (Sma1641), Sma1163 (P1B-5-ATPase), P1B-type ATPases and others, and 14 loci (gene annotation corresponds to Rlv 3841 genome) encoded enzymes responsible for the tolerance/resistance mechanism of Co/Ni, Pb, Cd, Cd/ZN, Ni, Ni/Fe, Cu/Zn, and Ni/Co, respectively. The genes are involved in HM detoxification through active efflux and sequestration [267,268]. Mutants with deletion of the resistance genes bbecome sensitive to the presence of corresponding HMs in the environment [38].

The rhizobium–legume interaction is a positive cooperation which aids growth of leguminous plants in various intractable conditions such as degraded, acidic and/or contaminated soils, particularly field soils with high HM concentration. The symbiotic process has evolved to demand a complex signal exchange between the legume and the rhizobial bacteria. In this process, the bacterial responses to various stresses such as low pH, oxidative stress, osmotic stress, and HMs play a significant role in symbiotic signaling pathways. The stress tolerance/resistance responses have been co-opted by the legume host and the bacterial symbionts to establish functional symbiosis [198], guarantee health growth, and most of all survive harsh living conditions.

## 4. Phytoremediation by Legume and Associated Rhizobia

### 4.1. Legume Growth, Nodulation, and Nitrogen Fixation under Heavy Metal Stress

Legumes colonized by NFB successfully compete with other non-legume plants in metal toxic sites due to their wide-ranging capacities for N fixation in the presence of HMs, which consequently promotes the root and shoot elongation and biomass of the crop [38,152,269] as well as rhizobial stress condition tolerance as mentioned above. Bai et al. [270] detected that all growth parameters and leaf catalase and peroxidase of plant species were significantly greater in a legume plant such as *R. pseudoacacia* L. than in non-legume ones such as *Rhus chinensis* Mill., which is typical phytoremediative non-legume species in soil environments polluted by HMs in China. Negative effect of the soil contamination on all growth parameters was significantly stronger in the non-legume than in the legume plant. The legume plant produced greater total plant, leaf, and root biomass. 

Plants with associated microorganisms growing in HM contaminated sites need to possess some level of tolerance/resistance to HM toxicity in order to survive. However, it should be mentioned that most rhizobia and legume plants are quite sensitive to excessive HMs such as Zn or Cd in field soils. For the metal non-tolerant legumes, toxic metals inhibit the nodule formation, the nodulation efficiency, and/or symbiotic N fixation [36,133,149,271]. Therefore, soil amendments, including excess limestone for making calcareous soil, and high phosphate and organic matter for improving field soil fertility, are imperative to reduce phytotoxicity in soil before legume planting and rhizobia inoculation [272]. Because there are no completely HM-tolerant WT legumes, Chaney et al. [273] proposed to introduce liming and fertilization steps to protect legumes hosts from phytotoxicity in the field soil making soil HMs less phytoavailable for the rhizobium–legume symbiosis. This procedure is likely to result in legumes and related rhizobia with sufficient inherent or selected HM resistance/tolerance to HMs to develop on phytostabilized soils contaminated with HMs [165,273]. Nevertheless, metal-resistant/tolerant rhizobia are isolated from native legumes, e.g., *Medicago, Trifolium, Viciae, Lotus*, and *Lupinus*, growing in HM contaminated field soils, e.g., Pb/Zn mine tailings (China) and Aznalcóllar pyrite mine (Spain) contaminated with 4.5 million m^3^ of slurries composed of acidic waters loaded with toxic metals and metalloids such as, Sb, Zn, Pb, Cu, Co, Tl, Bi, Cd, Ag, Hg and Se [29,169,269,274,275]. In all cases, isolated *Rhizobium* species formed effective normal-sized pink nodules on their host plant roots. Moreover, Carrasco et al. [274] while confirming the presence of the HM resistance genes in some *Rhizobium* strains by PCR and Southern blot hybridization, noticed that the first steps in nodule formation appear to be influenced more by HMs than by N_2_ binding. Therefore, another procedure for the implementation of phytoextraction strategies may be used through the identification of legumes and/or associated NFB species, to enable tolerating enormous quantities of HMs in the soil and absorb large amounts of at least one metal [143]. 

### 4.2. The Success of Biorhizoremediation of Heavy Metals

The success of phytoremediative plant species predominately depended not only on the tolerant/resistant properties, but also on fast plant growth rates, high biomass production, the ability of HM accumulation in plant biomass, and the ability to obtain N by symbiotic relationship with NFB [149,276,277,278,279]. The symbiosis between metal-tolerant/resistant NFB and legumes greatly influence the efficiency of phytoremediation through promoting plant growth, thus enhancing plant biomass under metal stress conditions [280,281]. Furthermore, highly biological N fixation can relieve the toxic effect of HMs, compensate the HM stress, and enhance survival and adaptation in the HM contaminated land in the reclamation process [282,283]. Thus, the use of legumes with associated NFB, presented in Figure 1, as a pioneering symbiotic plants for phytoremediation has the following benefits [284,285]: Improve the soil properties to allow other plants to grow through immobilization of contaminants, enhancing organic content, and modifying rhizosphere population;Increase diversity of microorganisms, especially rhizobacteria and arbuscular mycorrhiza fungi to improve and stabilize the ecology of the polluted and contaminated field soils;Provide additional nitrogen and phosphorus compounds to the field soil to improve its fertility and ability to support biological growth;Improve plant living conditions to allow legumes and other plants to grow in HM and other stress conditions;Promote plant growth.

Phytostabilization and phytoextraction are dominant applications of HM remediation by legumes [165]. Due to the participation of rhizobia, the uptake of metals by both roots and shoots increases with increasing biomass. Wherein, in the most tested plants, the enhancement of shoot biomass is much greater than the enhancement of root biomass and simultaneously the concentration of HMs in roots is still larger than in the shoots [36,142,144,149,159,286]. However, there are some *Lupinus* and *Astragalus* species that accumulate higher concentration of HMs in shoots than in roots [275,287]. It is particularly important to emphasize that the HM concentration in shoots of most of the legume plants used for biorhizoremediation is below the threshold allowed for animal grazing [133]. On the other hand, there are some HM hyperaccumulators among the legumes, such as *Lupinus* and *Astragalus* species. *Lupinus albus* L. growing in acidic soil can accumulate Zn in both shoots (748.3 mg/kg–3.61 g/kg) and roots (up to 4.65 g/kg), in a concentration exceeding the amount allowed for consumption by animals (500 mg/kg) [149,287,288]. Similarly, the accumulation of Cu by *Lupinus luteus* in both shoots (52.1 mg/kg) and roots (150.7 mg/kg) exceeds the Cu norm for animal consumption (40 mg/kg) [149,165]. Furthermore, some *Astragalus* species, e.g., *A. sinicus* Thunb. and *A. bisulcatus* Hook., hyperaccumulate Se (up to 6 g/kg in leaves and 1 g/kg in fruit and seeds) [275,289,290]. Consequently, these plants are not suitable for animal consumption, but they are useful for phytoremediation [165]. However, there are some reasonable and helpful disposal and utilization methods for getting rid of plant biomass containing HMs after phytoremediation [291]. 

Moreover, there are some legumes that can accumulate different metals in different plant parts at the same time, e.g., *Cytisus scoparius* L. growing in high metal contents accumulates Zn and Pb in its roots, Zn in the aerial part and excluding mostly Cd from its tissues. In this case, *C. scoparius* L. behaves like a Zn accumulator plant, while simultaneously, it also behaves like a Pb phytostabilizer and as a Cd excluder species [139]. Interestingly, Vázquez et al. [147] detected that the percentage of total Cd adsorbed by the cell wall of white lupin ranged from 29 to 47% in leaves, 38 to 51% in stems and from 26 to 42% in roots depending on the Cd supply. Cd induces the synthesis of elevated levels of PCs in lupin plants, mainly in roots, with PC3 being the major PC. The number of Cd complexed by thiols accounted for approximately 20% of the total Cd in leaves, 40% in stems and 20% in roots. Therefore, cell-wall retention could account for more than twice the amount of Cd complexed by PCs in leaves and roots. In stems, both mechanisms contributed equally to Cd detoxification. Thus, white lupin plants use cell-wall binding and then, the PC production, as effective mechanisms of Cd detoxification.

The symbiotic NFB support plant growth by adsorption and tolerance/resistance to HMs and toxic metalloids. NFB colonized nodules can enhance HM accumulation in root nodules, while non-symbiotic rhizobia living free in the rhizosphere may reduce HM toxicity in the contaminated field soils by biosorption, chelation, immobilization, and precipitation. Furthermore, upon the onset of legume–rhizobium symbiosis, the nodules can serve as HM buffer areas, which provide plants with additional storage space for HMs and reduce the risk of the plant being directly exposed to HMs [165]. Moreover, nodules with high NFB concentration may also be effective as biosorbents and storage for HMs in comparison with nodule-free plants. For example, the Cd, Cu and Pb uptake by *Mimosa punica* L. nodulated by *Cuprividus taiwanensis* from family *Burkholderiaceae* was 70%, 12% and 86%, respectively, higher than that of non-nodulated plants showing the effectiveness of using nodulated legumes for HM removal [12,165]. Interestingly, Saadani et al. [160] showed that legume inoculation significantly increased their biomass production, as well as N and P content. Moreover, in a crop rotation system, the use of NFB-legume symbiosis product as a green manure before cultivation of edible greens such as lettuce positively affected soil fertility due to a higher content of organic matter that quickly decomposed in the soil. The improvement of soil fertility increased lettuce yield and its N and P content. Furthermore, inoculated legumes extracted and accumulated more HM than non-inoculated legumes. The symbiosis between legume and NFB plays the role of organic trap for HMs, making HMs unavailable for following crops. The lettuce HM content, such as Zn and Cd, in edible parts was significantly decreased. The legume–NFB symbiosis enhances non-legume crop yield and limits HMs translocation to food chain [160]. Therefore, NFB inoculation improves the HM contaminated soil remediation efficiency of the legume plants, especially in the context of environmental protection.

In addition, NFB can improve both HM resistance and phytoextraction capacity of symbiotic plants [292]. For example, extracellular EPSs produced by NFB act as a first protective barrier which immobilizes HM ions from the cytoplasm, especially the loosely bound EPSs, which have a rough surface and a lot of honeycomb pores, promoting immobilization of HMs [293]. Additionally, there are the ion-selective ATPase pumps which conserve the HM transfer system in NFB [294,295,296]. NFB can also change the properties of root absorption, enhance the quantities of root exudates, and enhance the amount of plant growth regulator enzymes such as 1-aminocyclopropane-1-carboxylate (ACC) deaminase [150,297]. Moreover, NFB can also stimulate the plant to synthesize antioxidant enzymes in roots, such as catalase and superoxide dismutase (SOD), and peroxidase in shoots to clean up the deleterious effects of HMs and re-establish homeostatic conditions [163,267,298,299,300]. 

### 4.3. The Enhancement of Heavy Metal Stress Response by other PGPB in Addition to the Legume–Rhizobium Symbiosis

It is important to mention that legume-based phytoremediation can also be improved by inoculation with a consortium of metal-tolerant/resistant NFB and other PGP rhizobacteria assisting phytoextraction of HMs with well-documented PGP activity such as *Actinomycetes* sp., *Azotobacter* sp., *Arthrobacter* sp., *Bacillus* sp., *Enterobacter* sp., *Leifsonia* sp., *Klebsiella* sp., *Kocuria* sp., *Proteus* sp., *Pseudomonas* sp., *Serratia* sp., *Sporosarcina* sp. and *Thlaspi* sp. [30,116]. For example, co-inoculation of *M. lupina* L. with *S. meliloti* (CCNWSX0020 strain) and *Pseudomonas putida* (UW4) resulted in larger plant biomass and higher total Cu accumulation than single inoculation with NFB [150]. Similarly, co-inoculation of *V. faba* L., *Lens culinaris* Medik., *Sulla coronaria* L., and *Lablab purpureus* L. with consortia of NFB and non-NFB such as *Bradyrhizobium* sp. (750 strain), *Pseudomonas* sp. (Az13) and *Ochrobactrum cytisi* (Azn6) are also effective in improving plant growth, pod yield, and accumulating more HMs in comparison with non-inoculated ones when grown in HM stress conditions [37,149,151]. In another study, Cr-tolerant rhizobacteria were isolated from the Cr contaminated rhizosphere area. The rhizobacteria were used as additional inoculating microorganism with *Bradyrhizobium* sp. vigna for *Vigna radiata* L. in Cr polluted soil. The inoculated legumes were found to exhibit an increase in biomass, root and shoot length compared to non-inoculated ones grown in the same conditions. Moreover, the inoculated legumes had a significant increase in Cd, Cr, Fe, Mn, Ni, Pb, and Zn accumulation in comparison with non-inoculated legumes [301,302]. Similarly, co-inoculation of *R. pseudoacacia* L. by non-NFB such as Variovorax sp., *Streptomyces* sp., and *Rhodococcus* sp. with Mesorhizobium loti enhanced phytoremediation in Cd, Zn and Pb contaminated soil [303].

The production of auxins such as IAA, cytokines, and gibberellins by many PGPB exerts additional significant effects on root growth and its architecture [304]. Furthermore, non-nitrogen fixing PGPB which have HM detoxification potential, produce, and secrete MBPs such as PCs, MTs, Cd-binding peptides, cysteines (gcgcpcgcg), and histidines (ghhphg). MBPs increase the accumulation and tolerance of HMs by PGPB through the synthesis of metal-binding proteins [116]. It is worth noting that besides bacteria, MBPs are also produced by some fungi such as yeast *Candida glabrata*, and *Schizosaccharomyces pombe*, and red mold *Neurospora crassa* [116]. These yeast and molds as well as *Saccharomyces cerevisiae, Aspergillus* sp., *Gloeophyllum sepiarium and Rhizopus oryzae* are used in HM remediation of contaminated sites [173]. Therefore, the improving plant–microbe interaction and introducing both useful rhizospheric bacteria and fungi are essential to increase biomass production and HM plant tolerance [302]. NFB and other PGP rhizobacteria act together as a community both within the root nodule and rhizosphere to succor health and survival of plants in HM contaminated soils. The bacteria and the legumes cooperate in the effective removal of HMs from the soil. 

## 5. Genetic Engineering in Improvement of Leguminous Plants and Their Rhizobial Partners for Phytoremediation

### 5.1. General Strategies of Plant Transgenesis for Increasing HM Tolerance and Accumulation

Phytoremediation shows great promise for a cleanup of HM polluted soil; however, several drawbacks, such as the slow growth of hyperaccumulators or the low metal accumulation ability of high biomass plants should be addressed. In addition to the exploration of plant–microbe interactions or nanotechnology, the solution may lie in genetic engineering of plants and their microsymbionts toward optimization of the metal remediation potential [305]. So far, many transgenic plants have successfully improved phytoremediation efficiency, including legumes (Table 3) [306,307]. Transgenesis consists in insertion of gene(s) from another organism, even distantly related, into the genome of recipient cells. It allows modification of plants with desirable traits that are not observed in the original line in a relatively short time. Hence, it seems an attractive prospect in the attempt to increase HM tolerance and accumulation in fast-growing plants, as engineering hyperaccumulators towards production of higher biomass seems to be less applicable. Transgenic plants intended for phytoremediation can be generated by nuclear or chloroplast transformation. The latter strategy is relatively rare; however, it has been successfully applied, as in the case of stimulation of phytoremediation of Hg by chloroplast-transgenic tobacco [308]. The advantages of chloroplast transformation include reduction in the gene silencing effect, avoidance of optimization of codons of bacterial transgenes, and limitation of the risk of gene transfer to the environment [309]. 

The success of plants in phytoremediation depends on their capability of metal uptake, translocation from the root to the shoot or other parts, and sequestration in specific cell compartments. Thus, genes coding for metal transporters in plants, such as ZRT/IRT-related proteins (ZIP), ATP-binding cassette transporters (ABC), or cation diffusion facilitators (CDF), are widely used for enhancement of metal accumulation [310,311]. Research has demonstrated that overexpression of genes coding for metal transporters in plants can significantly improve metal accumulation (e.g., [312,313,314,315]). In turn, the capacity of plants to cope with HM toxicity is facilitated by detoxification of excess metals entering cell cytosol through complexation by metal ligands, compartmentation, and antioxidant activity. Hence, studies focused on enhancement of metal tolerance typically involve overexpression of metal-binding proteins, especially MTs and PCs (e.g., [316,317,318]). MTs, which are widespread in nature, display high affinity for metal ions due to the presence of clusters of cysteine residues having thiol groups [319]. In plants, MTs are divided into 4 classes; each of them has a different cysteine composition and plays a specific function in the cell. In turn, PCs are metal-binding peptides that are 5–17 amino-acid-long and they are synthesized from glutathione [320]. Due to the variety of their unique structures, they offer higher metal specificity and affinity than MTs. This explains the interest in the use of PC-encoding genes in improving HM stress tolerance in plants. However, the presence of ɤ bonds in PCs makes their biosynthesis dependent on the activity of specific enzymes, with phytochelatin synthetase (PCS) and c-glutamyl cysteine synthetase (c-GCS) as the key enzymes; thus, genes coding for these enzymes are common targets for genetic manipulation. Another common strategy of plant transgenesis for bioremediation is overexpression of genes encoding antioxidant enzymes and ROS scavengers, with most promising ATP-sulfurylase (ATPS), ascorbate peroxidase (APX), and glutathione S-transferase (GST) [321,322,323]. 

Other genes, which are less commonly harnessed to improve phytoremediation, are those related to DNA repair and expression, as in the case of *M. truncatula* Gaertn. overexpressing the DNA repair gene *MtTdp2α* with an attenuated Cu effect and potato overexpressing the StDREB transcription factor with increased tolerance to Cd and Cu [324,325]. In addition, the latest reports show a possibility of modulating the epigenetic regulation of HM-responsive genes for stimulation of HM tolerance in plants [311]. For example, overexpression of the SlRING1 gene, coding for ubiquitin ligase, resulted in enhanced tolerance to Cd in tomato plants by limiting Cd accumulation and alleviating oxidative stress [326]. Gene stacking refers to combining and simultaneous expression of multiple genes into a host plant. Compared to single-gene transgenic plants, dual-gene plant mutants often showed significantly higher HM tolerance and accumulation, as in the case of *A. thaliana* L. overexpressing genes coding for PC and GST as well as *Brassica juncea* L. (Indian mustard) overexpressing genes coding for selenocysteine methyltransferase and ATPS [327,328]. 

*Medicago truncatula* Gaertn. is a model legume plant used in investigations of root symbiotic associations and abiotic stress tolerance [329]. *Medicago* spp., which are most often subjected to transgenesis aimed at increasing the phytoremediation potential, exhibit high tolerance to various HMs, as they are able to accumulate metals in roots. Importantly, the convenient procedure for *Agrobacterium*-mediated transformation yields stable fully transgenic plants or composite plants, in which only roots are transformed while the aerial part is genetically unchanged [330,331]. Generation of composite legumes is sometimes preferred as a way for limitation of transgene spread in the environment and prevention of translocation of metals from roots to shoots, which is important in phytostabilization. Root-specific expression of introduced genes can also be obtained using nodulation-specific promoters, e.g., *nifH* [332]. 

**Table 3 biology-11-00676-t003:** Transgenic leguminous plants for improving of a heavy metal phytoremediation efficiency.

Gene (s)	Gene Function	Gene Origin	Gene Host	Effect Comparing to the Wild-Type (WT) Control	References
*GST* and *CYP2E1*	Glutathione S-transferase Cytochrome 450 2E1	*Human*	*Medicago sativa* L.	Improved tolerance of plants towards Cd/trichloroethylene (TCE) mixture	[333]
*GST* and *CYP2E1*	Glutathione S-transferase Cytochrome 450 2E1	*Human*	*M. sativa* L.	Improved tolerance of plants towards Hg/TCE mixture. Ameliorated plant growth with a longer root system enhanced accumulation of pullulans	[334]
*MtTdp2α*	Tyrosyl-DNA phosphodiesterase II	*Medicago truncatula* Gaertn.	*M. truncatula* Gaertn.	Improved Cu tolerance. Reduction in necrosis volume, decrease in double strand breaks in DNA	[324]
*mt4a*	Metallothionein	*Arabidopsis thaliana* L.	*M. truncatula* (composite plants)	Increased Cu tolerance, reduction in oxidative stress. Improved nodulation	[119]
*mt4a*	Metallothionein	*A. thaliana* L.	*M. truncatula* (composite plants)	Enhanced Cu tolerance, reduction in oxidative stress. Improved plant growth and nodulation	[332]
*ATPS*	ATP sulfurylase	*A. thaliana* L.	*M. sativa* L.	Increased tolerance towards a mixture of Cd, Ni, W, Cu, and Pb. Enhanced metal uptake and accumulation in roots and shoots	[322]

### 5.2. Transgenesis of Rhizobia for Improving Phytoremediation in Symbiosis with Legumes

Molecular mechanisms and genetic determinants of metal tolerance in rhizobia have been extensively studied [38]. Rhizobia are generally more tolerant to HMs than their legume partners; however, the use of these metal-binding bacteria alone for bioremediation does not prevent recycling of toxic metals to the soil. Instead, rhizobia with improved traits associated with metal uptake and plant growth promotion are regarded as drivers in stimulation of phytoremediation in legumes [285,335]. The use of legumes in association with transgenic rhizobia for phytoremediation was initially called “symbiotic engineering” [336]. This approach, using the advantages of both leguminous plant and their microsymbionts, allows further improvement of HM uptake and tolerance as well as symbiotic performance of partners, not achieved by using selected bacteria living in nature (Table 4). A pioneer experiment in this field was performed by Sriprang et al. [337], who introduced an *Arabidopsis* gene coding for MT (*MTL4*) to the genome of *Mesorhizobium huakuii* subsp. *rengei* B3 under the control of the *nifH* and *nolB* promoters. The expression of *MTL4* increased the accumulation of Cd both in free-living cells (in low oxygen conditions) and in *Astragalus sinicus* Thunb. nodules (1.7–2.0-fold). It was calculated that 1 nodule absorbed up to 1.4 nmol Cd from the polluted soil; however, the different concentrations of this metal did not directly influence the accumulation level. Next, a gene from *Arabidopsis thaliana* L. coding for PCS (*PCS_At_*) was fused with the *nifH* promoter and introduced to the *Mesorhizobium huakuii* B3 strain. The expression of the transgene increased the ability of free-living bacteria to accumulate Cd in a range from 9- to 19-fold; however, in a symbiotic relationship with *A. sinicus* Thunb., the transgenic rhizobia increased Cd accumulation approx. 1.5-fold, compared to the non-transformed control [338]. Overexpression of both *PCS_At_* and *MTL4* genes in *M. huakuii* B3 resulted in additive accumulation of Cu in free-living cells, with an up to 25-fold increase in its concentration. Nevertheless, the rise in the Cu content in the *A. sinicus* roots was up to three-fold higher compared to the control native bacteria, suggesting that the accumulation of this metal is limited by other factors [339]. In addition, it was demonstrated that the presence of the multicopy *PCS_At_* gene in free-living *M. huakuii* B3 cells resulted in to an approx. seven-fold increase in Cd accumulation, compared to these possessing only a single copy [338]. Recently, studies on Cd phytoremediation by a legume–rhizobium system, where both partners contained transgenes, were carried out by Tsyganow et al. [340]. They used two *Rhizobium leguminosarum* bv. *viciae* strains expressing different pea MT encoding genes (*PsMT1* and *PsMT2*) for inoculation of a pea (*Pisum sativum* L.) mutant with increased Cu tolerance and accumulation or WT plants as a control. With and without Cd treatment, both transgenic strains differed in their effect on plant growth, Cd content, and histological organization of nodules, which suggests a specific expression pattern and function of the introduced genes. Generally, the strain containing *PsMT1* was more efficient in increasing plant biomass with the native pea and limiting Cd content in shoots of both native and mutant plants. On the other hand, the plant inoculated with the *PsMT2* containing strain accumulated the highest amount of this metal in the plant roots, which is promising for probable future application. 

Transgenic rhizobia have also been generated for increasing Cu phytostabilization in legumes [341]. Cu resistance genes from *Pseudomonas fluorescens* (*copAB*), under the control of the *nifH* promoter, were overexpressed in *Ensifer medicae*. The recombinant strain in symbiotic relationships with *M. truncatula* Gaertn. alleviated the toxic effect generated by Cu at a moderate concentration and accumulated two-fold higher amounts of Cu in the root nodules than the native strain. In addition, the Cu content in the plant roots was significantly higher than in the shoots, which makes the plasmid containing *nifH*/*copAB* genes a promising tool for improving phytostabilization. The same plasmid was also used by Pajuello et al. [342] in a double genetically modified symbiotic system. The composite *M. truncatula* Gaertn plants expressing an *Arabidopsis* gene encoding MT (*mt4a)* exhibited higher tolerance towards Cu with reduced oxidative stress and enhanced nodulation, compared to native plants. Inoculation of these with *E. medicae* expressing *copAB* genes had an additive phytostabilization effect, enhancing Cu accumulation in roots without its increase in shoots. Similar experiments, in which both symbiotic partners were genetically modified for improvement of Cu phytostabilization by the legume/rhizobium system was also carried out by Pérez-Palacios et al. [332]. 

Symbiotic relationships of the red clover *Trifolium pretense* L. with recombinant *R. leguminosarum* expressing the S-adenosylmethionine methyltransferase gene (*CrarsM*) as a novel approach to arsenic bioremediation was investigated by Zhang et al. [343]. The engineered rhizobial strains had capability of methylating As in both free-living and symbiotic state. In fact, the products of arsenic methylation are generally less toxic than inorganic As, and some of them can undergo volatilization; however, higher plants do not have the ability to carry out this reaction, while As methylation by indigenous microbes in soil is limited. Therefore, As methylation by rhizobia in nodules, which are specific compartments providing their high survival capability, constitute a promising way for improvement of arsenic phytoremediation in leguminous plants.

**Table 4 biology-11-00676-t004:** Summary of studies using transgenic rhizobia in association with leguminous plants for heavy metal phytoremediation.

Gene(s)	Genome Function	Gene Origin	Gene Host	Legume Partner	Effect Comparing to the Wild-Type (WT) Control	References
*MTL4*	Metallothionein	*Arabidopsis thaliana* L.	*Mesorhizobium huakuii* subsp. *rengei* B3	*Astragalus sinicus* Thunb.	Increased Cd accumulation in free-living cells and nodules	[337]
*PCs*	Phytochelatin synthase	*A. thaliana* L.	*Mesorhizobium huakuii* subsp. *rengei* B3	*A. sinicus* Thunb.	Increased Cd accumulation in free-living cells and nodules	[338]
*MTL4* and/or *AtPCS*	Metallothionein Phytochelatin synthase	*A. thaliana* L.	*Mesorhizobium huakuii* subsp. *rengei* B3	*A. sinicus* Thunb.	Increased Cd accumulation in free-living cells and root nodules in hydroponic culture. Additive effect in the case of expression of both genes. Increased Cd accumulation in nodules and roots of *A. sinicus* in rice paddy soil	[339]
*PsMT1*	Metallothionein	*Pisum sativum* L.	*Rhizobium leguminosarum* bv. *viciae*	*Pisum sativum* SGE (WT) or *Pisum sativum* SGECd^t^ (mutant with increased tolerance to Cd)	Positive effect on WT plant biomass during growth in the presence of Cd. Reduction in the Cd content in shoots of both WT and mutant plants, proper organization of nodules	[340]
*PSMT2*	Metallothionein	*P. sativum* L.	*R. leguminosarum* bv. *viciae*	*Pisum sativum* SGE (WT) or *P. sativum* SGECd^t^ (mutant with increased tolerance to Cd)	Increased Cd accumulation in mutant plants	[340]
*copAB*	Cu^+^-ATPase	*Pseudomonas fluorescens*	*Ensifer medicae* MA11	*Medicago truncatula* L.	Alleviation of the toxic effect of Cu in plants. Increased plant growth, nodule numbers, nitrogen contents, and photosynthetic efficiency. Increased Cu accumulation in nodules and roots with decreased accumulation in shoots	[341]
*copAB*	Copper resistance proteins; Cu^+^-ATPase	*P. fluorescens*	*E. medicae* MA11	*M. truncatula* composite mutant expressing *mt4a* gene coding metallothionein	Enhanced Cu tolerance in plants. Improved Cu accumulation in roots, diminished Cu translocation from roots to shoots	[332]
*CrarsM*	S-adenosyl methionine methyl- transferase	*Chlamydomonas reinhardtii*	*Rhizobium leguminosarum* bv. *trifolii* R3	*Trifolium pratense* L.	Capability to methylate As by free-living bacteria and in symbiosis with plants. Methylated As forms detected both in nodules and shoots. Lesser amounts of methylated As species were volatilized.	[343]

### 5.3. Other Genetic Engineering Methods to Be Used for Improving Bioremediation in Legumes

An alternative technology to the conventional transgenesis is cisgenesis or intragenesis, which has been developed for improvement of crops to overcome the safety concerns about the presence of foreign DNA [344]. Cisgenesis is based on introduction of unchanged genes and their regulatory elements from the same species as the host, while intragenesis refers to a new combination of genes and regulatory elements from the species pangenome. These techniques are regarded to be valuable, especially in increasing the production of biomass of natural metal hyperaccumulators; however, their application in plants is still limited, as extensive knowledge of the genetics of modified plants is required. However, a cis-hybrid *E. meliloti* strain has recently been created via transfer of a symbiotic megaplasmid pSymA from a donor of the same species. The hybrid strain showed a phenotype consistent with the parental strains, proving the feasibility of the large-scale genome modification of bacteria for maximization of biotechnological applications [345]. In fact, genomes with multipartite structures such as those present in rhizobia are particularly interesting for use in cisgenesis; however, many functional dependencies between co-adapted replicons should be taken under consideration [346,347]. 

Introduction of natural resistance plasmids from a highly resistant host to its close relatives exhibiting high symbiotic performance is usually mentioned as another promising strategy of increasing HM tolerance and accumulation in legumes. A suitable candidate for creation of novel bacterial strains for use in As bioremediation could be the pSinA plasmid derived from highly metallotolerant non-symbiotic *Ensifer* sp. Analyses have revealed that pSinA encodes gene clusters for arsenic resistance and arsenite oxidation (*ars* and *aio* genes, respectively), and transfer of this broad-range self-replicon via conjugation to other bacteria resulted in acquisition of HM resistance and As oxidation traits [348].

Enhancement of the phytoremediation potential of plants can also be conducted by gene silencing. It is a conserved defense mechanism that knocks down the gene expression based on RNA interference, which is mediated by siRNA and miRNA [349]. In contrast to native plants, *Arabidopsis* plants with a silenced gene encoding arsenate reductase showed the ability to translocate As from roots to shoots, which enhanced their capability of this metal accumulation [350]. Similarly, increased Cd translocation to shoots was caused by silencing a gene coding for the OsNRAMP5Cd transporter in rice [351]. To date, there have been no reports on the application of gene silencing for enhancement of bioremediation in legumes. However, Zhou et al. [352] have reported identification of several miRNAs involved in the response to HM stress of *M. truncatula*. 

With the advent of the genome editing technology CRISPR/Cas9 (clustered regularly interspaced palindromic repeats) facilitating highly specific gene knockouts, site-directed gene insertion, or gene expression modulation, improvement of plant crops has gained a new perspective [353]. Despite some limitations in using the CRISPR/Cas9 technology in plants, especially the high lethality of Cas9, it is generally accepted that it will accelerate genetic manipulation of plants to increase the efficiency of the legume/rhizobium system for phytoremediation by customizing desired improvements, especially when the classical gene transfer or overexpression method fails [134,311]. It was also suggested that implementation of the CRISPR/Cas9 system in rhizobia can improve legume/rhizobium interactions leading to stimulation of plant growth, e.g., by targeting genes responsible for overcoming of the ethylene effect inhibiting root growth [335]. So far, the CRISPR/Cas9 system has been successfully used for modulation of natural metal accumulating crop plants toward production of safe crops growing on contaminated soil. For example, it was demonstrated that knocking out the metal transporter gene *OsNramp5* with the use of the CRISPR/Cas9 system yielded rice lines with low Cd content without affecting the yield [354].

## 6. Conclusions and Further Perspective

Legumes with associated rhizobia are an elite group for novel and effective soil restoration, including mine lands, decreasing the mobility of heavy metals and preventing their dispersion to another ecosystem compartments. Legumes and rhizobia interact with each other in various complex manners, including metal-tolerance or metal-resistance mechanisms, plant growth-promoting properties, nitrogen fixing ability, phytohormone production, and phosphorus solubilization. These rhizobium traits most likely with rhizosphere PGPB and mycorrhizal fungi promote legume growth in heavy metal contaminated soils by increasing legume yields, heavy metal accumulation in legumes, nitrogen and phosphorus content in both legumes and contaminated lands. 

Moreover, genetic engineering is a powerful tool for development of plants with desirable traits facilitating an effective clean-up of contaminated soils, especially given the development of innovative technologies. Transgenesis is the most exploited technique; however, some results of introduction of foreign genes, especially these encoding the antioxidant machinery, are rather contradictory and confusing, pointing out morphological and physiological alterations in transgenic plants or a lack of effects in metal tolerance. Additionally, transgenic plants are usually regarded as a danger for ecosystems due to their potential of outcrossing with wild relatives or uncontrolled spread. Therefore, their application, even only in field tests, is limited in some countries. This has accelerated the development of new more acceptable techniques, such as cisgenesis. Recently, the plant genome editing approach has received considerable attention from biotechnologists. As a highly precise method of genetic engineering, gene editing results in tiny changes in the genome, which are indistinguishable from naturally occurring mutations. Therefore, there is a hope that plants with improved genomes will not be associated with GMOs and will not be subject to the same stringent regulations as GMOs. Understanding the gene functions and dissection of the signaling network responsible for ameliorating the toxic effect of HMs is a determining factor in the genetic engineering approach. It has been confirmed that the mechanisms of HM detoxification and accumulation in plants involve a high number of different genes, which may be regarded as the main limitation of the genetic engineering approach. Nevertheless, technological breakthroughs involving modern multi-omics analysis and gene editing methods will provide better understanding the molecular mechanisms underlying the process of phytoremediation. 

## Figures and Tables

**Figure 1 biology-11-00676-f001:**
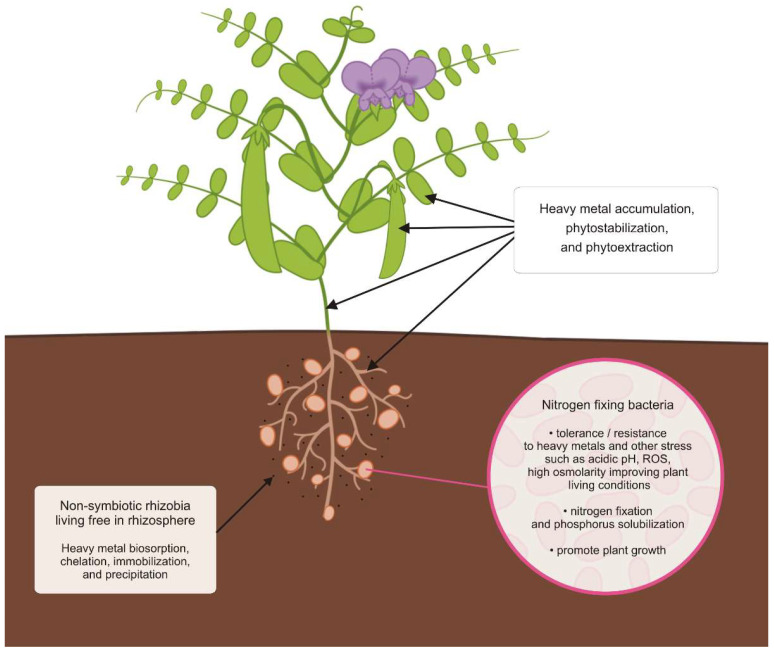
Legume–rhizobium symbiosis involved in the biorhizoremediation process for heavy metal removal.

**Table 1 biology-11-00676-t001:** The toxic effects of excess heavy metals on plants.

Heavy Metals	The Toxic Effects of Heavy Metals on Plants	References
Cd	Reduction in biomass and root length; inhibition of seed germination; growth reduction; wilting; chlorosis, and cell damage	[56,57,78,79,80,81]
Cu	Inhibition of root, shoot, and leaf development; quantity reduction in leaves per plant; decreased antioxidant activities; shoot length reduction; decreased total chlorophyll content; reduction in chlorophyl biosynthesis; decreased enzyme activities; decreased plant growth and yield; leaf chlorosis; generation of oxidative stress and ROS	[56,57,77,79,82,83,84,85,86]
Zn	Decreased total chlorophyll content; reduction in transpiration rate, inhibition of transport of microelements; limitation of root and shoot growth; reduction in photosynthetic and respiratory rate; enhancement of generation of reactive oxygen species; chlorosis in the younger leaves; reduction in germination	[56,57,79,87,88,89,90]
As	Inhibition of growth and low crop production; reduction in leaf quantities; chlorosis; leaf senescence necrosis; defoliation; reduction in leaf area and dry matter production; reduction in shoot and root growth; restricted stomatal conductance and nutrient uptake; chlorophyll degradation; limited biomass and yield production; overproduction of reactive oxygen species (ROS) leading to carbohydrate, protein, and DNA damage.	[57,91,92,93,94]
Ni	Reduction in chlorophyll content; decreased levels of sugar, starch, and protein nitrogen; decrease in shoot yield; chlorosis; inhibition of root growth; inhibition of growth, induction of chlorosis, necrosis, and wilting; generation of ROS	[57,79,95,96,97]
Pb	DNA damage; decrease in chlorophyll content; decrease in protein content; stunted foliage; reduction in photosynthesis; impaired nutrient uptake; decrease in seed germination, root elongation, decreased biomass; inhibition of chlorophyll biosynthesis; inhibition of mineral nutrition and enzymatic reactions, induction of ROS production	[42,56,57,98,99,100]
Cr	Inhibition of root, steam, and leaf growth; inhibition of chlorophyll biosynthesis; induction of oxidative stress; inhibition of photosynthesis; inhibition of seed germination and seedling development; reduction in root and shoot biomass, quality of flowers, and crop yield	[57,77,101,102,103]
Co	Inhibition of plant growth; chlorosis in young leaves; reduction in biomass; inhibition of greening	[57,76,104,105]
Fe	Reduction in root and shoot growth; hindered growth, reduction in chlorophyll content in older leaves, decreased sugars, starch, and protein nitrogen contents	[106,107]
Mn	Reduction in biomass production; adversely affects nutrient uptake; hindered seedling growth; induction of oxidative stress	[108,109]

**Table 2 biology-11-00676-t002:** Selected wild-type legume plants used for biorhizoremediation.

Symbiotic Species		Heavy Metal	References
Legume	Rhizobium		
*Anthyllis vulneraria* L.	*Mesorhizobium* sp.	Zn, Cd, Pb	[136,137]
*Astragalus thaliana* L.	-	Cd	[138]
*Cytisus scoparius* (L.) Link	-	Pb, Zn	[139]
*Glycine max* (L.) Merr	-	Cd	[140]
*G. max* (L.) Merr	*Bradyrhizobium japonicum*	As	[141]
*Lablab purpurens*	*Rhizobium* sp.	Cd, Cu, Zn	[142]
*Lathyrus sativus* L.	-	Pb	[143]
*Lens culinaris* Medik.	*Rhizobium leguminosarum*	Zn, Cu, Cd, Pb	[144,145]
*Lotus corniculatus* L.	*Bradyrhizobium liaoningense*	Zn, Pb	[146]
*Lupinus albus* L.	-	Cd, As, Cu, Pb, Zn	[147,148]
*Lupinus luteus* L.	*Bradyrhizobium* sp.	Cd, Cu, Pb, Zn	[149]
*Lupinus* sp.	*Bradyrhizobium* sp.	Cd, Cu, Pb, Zn	[133]
*Medicago lupulina* L	*Sinorhizobium meliloti*	Cu	[150]
*Medicago sativa* L.	*Rhizobium leguminosarum* bv. *trifolii*	Cr, Cu, Zn, Hg	[151,152]
*M. sativa* L.	*Sinorhizobium meliloti*	Cu	[153]
*Pisum sativum* L.	*Rhizobium* sp.	Cd, Cu, Zn	[154,155]
*Prosopis laevigata* (Willd.) M.C. Johnst.	-	Cu, Pb, Zn	[156]
*Robinia pseudoacacia* L.	-	Pb, Cu, Zn	[38,157,158]
*Sesbania rostrata* Bremek. & Oberm.	-	Cu, Pb, Zn	[159]
*Sesbania sesban* L.	-	Cu, Pb, Zn	[159]
*Vicia faba* L.	-	Pb, Zn, As, Cd, Cu	[160,161]
*V. faba* L.	*Rhizobium leguminosarum*	Cd, Cu, PB	[162]
*Vicia sativa* L.	-	Cd, Pb, Zn	[161]

- data not available.

## Data Availability

Not applicable.

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
