# Peer review of "Utilization of Legume-Nodule Bacterial Symbiosis in Phytoremediation of Heavy Metal-Contaminated Soils"

_biology, 2022, doi:10.3390/biology11050676_

Round 1

Reviewer 1 Report

The review "Legume symbiotic nitrogen fixation by nodule bacteria in phytoremediation of heavy metals" aims to provide a comprehensive description of the main effects of metal contaminants in nitrogen-fixing leguminous plants". The manuscript is very clear and brings important informations regarding the subject. The description of the current genetic strategies to increase the capacity of plants and bacteria to accumulate heavy metal was the highlight of the manuscript. 

I do recommend the review publication after minor revision. 

Please take in consideration the suggestions/recommendations mentioned below: 

Line 15: At Simple Summary, please delete the comma after "extraction or stabilization".

Lines 15 - 17: Please revise the sentence avoiding repetition of the word "

directly".

Line 32: "Wild-type" is misspelled. Please correct it.

Line 32: When authors say "...benefits of using the legume-rhizobia symbiosis with both wide-type and genetically modified to enhance an efficient recovery of contaminated lands." what do they mean? Genetically modified isolates, strains, bacteria, plants...?

Line 112: "...legumes can introduced in modern cropping systems". It seems the word "be" is missing. Please revise the sentence.

Lines 238 and 239: Authors said "The use of legume plants in phytoremediation of metal-contaminated soils represents a new area of research..." Although it is a very important area, I do not agree this research area is new since some works mentioned by authors in table 02 were published in 2006, 16 years ago!

Line 245: Table 02 subtitle. Please replace "wide-type" to "wild.type".

Line 248-250: Please provide more information justifying why those plants could be used as potential phytoremediation species.

References:

The year of several references is not in bold.

Author Response

We appreciate Reviewer’s suggestions. They are unbelievably valuable for our manuscript.

According to recommendations of Reviewer #1 we revised our manuscript to meet all requirements:

Reviewer’s comments:

Line 15: At Simple Summary, please delete the comma after "extraction or stabilization".

We removed the comma after stabilization.

Lines 15 - 17: Please revise the sentence avoiding repetition of the word " directly".

We removed one world “directly” from this sentence.

Line 32: "Wild-type" is misspelled. Please correct it.

We corrected the formula from “Wide type” to “Wild type” in the entire manuscript.

Line 32: When authors say "...benefits of using the legume-rhizobia symbiosis with both wide-type and genetically modified to enhance an efficient recovery of contaminated lands." what do they mean? Genetically modified isolates, strains, bacteria, plants...?

We corrected this part as follows:  “…both wild type and genetically modified plants and bacteria…”

Line 112: "...legumes can introduced in modern cropping systems". It seems the word "be" is missing. Please revise the sentence.

In line 137, we added the word “be” in the sentence.

Lines 238 and 239: Authors said "The use of legume plants in phytoremediation of metal-contaminated soils represents a new area of research..." Although it is a very important area, I do not agree this research area is new since some works mentioned by authors in table 02 were published in 2006, 16 years ago!

In line 264, we corrected the part as follows: “… represents a very important area of research…”

Line 245: Table 02 subtitle. Please replace "wide-type" to "wild.type".

In line 270, we replaced “wide-type” to “wild type” in table 02 subtitle.

Line 248-250: Please provide more information justifying why those plants could be used as potential phytoremediation species.

To provide more information justifying why the plants could be used as potential phytoremediation species, between lines 277 and 279, we added a following sentence: “These selected tree species showed favorable uptake of HMs, translocation, and HM storage in different plant parts as accumulators, without the addition of chelating agents and/ or organic adjustment.”

References: The year of several references is not in bold.

We corrected the format of all references according to Author Guidelines. According to the Guidelines, books are to have unbolded years of publication.

Moreover, the native speaker checked English language and style.

Reviewer 2 Report

The title needs to be revised, such as “Utilization of legume-nodule bacteria symbiosis in phytoremediation of heavy metals-contaminated soils”

An outline or table of content is necessary for such a long review.

The authors set up the subheading 3.4.1, but no subsequent subheadings followed.

Part 3 should focus the fitness of legumes aided by rhizobia, and Part 4 should focus on the efficiency of legume-rhizobia for remediation of HM contaminated soils.

Author Response

List of responses to each of the Reviewer #2 comments

We appreciate Reviewer’s suggestions. They are very valuable for our manuscript.

According to recommendations of Reviewer #2 we revised our manuscript to meet all requirements:

The title needs to be revised, such as “Utilization of legume-nodule bacteria symbiosis in phytoremediation of heavy metals-contaminated soils”

We changed the title as follows: “Utilization of legume-nodule bacterial symbiosis in phytoremediation of heavy metal-contaminated soils.” The native speaker corrected the title.

An outline or table of content is necessary for such a long review.

Between lines 37 and 60, we added table of content.

The authors set up the subheading 3.4.1, but no subsequent subheadings followed.

In line 440, we changed the subheading from 3.4.1 to 3.5.

Part 3 should focus the fitness of legumes aided by rhizobia, and Part 4 should focus on the efficiency of legume-rhizobia for remediation of HM contaminated soils.

In part 3 between lines 502 and 510, we added a following paragraph which briefly summarizes the fitness of legumes aided by rhizobia: “The rhizobium-legume interaction is a positive cooperation which aids growth of leguminous plants in various intractable conditions such as degraded, acidic and / or contaminated soils, particularly field soils with high HM concentration. The symbiotic process has evolved to demand a complex signal exchange between the legume and the rhizobial bacteria. In this process, the bacterial responses to various stresses such as low pH, oxidative stress, osmotic stress, and HMs play a significant role in symbiotic signaling pathways. The stress tolerance / resistance responses have been co-opted by the legume host and the bacterial symbionts to establish functional symbiosis [198], guarantee health growth, and most of all survive harsh living conditions.”

In part 4 between lines 630 and 632, we added a following part which emphasize the efficiency of legume-rhizobia for remediation of HM contaminated soils: “Therefore, NFB inoculation improves the HM contaminated soil remediation efficiency of the legume plants, especially in the context of environmental protection.” Furthermore, between lines 680 and 683, we also added summarizing sentences: “NFB and other PGP rhizobacteria act together as a community both within the root nodule and rhizosphere to succor health and survival of plants in HM contaminated soils. The bacteria and the legumes cooperate in the effective removal of HMs from the soil.”

Reviewer 3 Report

Minor corrections should be made in several sentences as follows:

Line 16: Is one of the two words "directly" redundant?

Rhizobia increase directly phytoremediation directly by nitrogen fixation ...

Line 32:  What did the authors mean by the following: wide-type or wild-type?

... both wide-type and genetically modified ...

Line 112: Was "Be" missing in the next sentence:  legumes can [be] introduced

Line 127-128: Was "comma or dot" missing at the end of the sentence?

The nature of pollution is quite varied

Line 213-214: Please check the meaning of the sentence. It does not seem to be created correctly.

Perhaps the authors meant the following:

Activation of the MT4 gene causes an increase in tolerance to metal ions, both Cu and Zn, in the leaves and roots of the host plant.

Lines 269, 609, 633, 635, 637, 842: Did you meant "PGPR"? There is no "PGPB" in the list of abbreviations. 

It should be replaced by " PGPR" everywhere or "PGPB" should be added to abbreviations.

Line 396: The word "as" is missing: act [as] a symbiotic signal

Line 401: Check for possible correction: "for acidic pH tolerance"  instead of     "for tolerance of acidic pH"

Line 443: correct: "stimulating effect"   instead of    " stimuling effect"

Line 448: Is the word "to" missing: "related [to] the high production"

Line 457: Correct: "species of Rhizobium increase"   instead of    "species of Rhizobium increases"

Line 487:  Comma must be removed:  "total, leaf"

Line 499-501:  The initial sentence is difficult to understant. It needs to be corrected, for example, as follows: "This procedure is likely to result in legumes and related rhizobia with sufficient inherent or selected resistance/tolerance to HMs to develop on phytostabilized soils contaminated with HMs."

Line 549-550: Need to be corrected: "On the other hand, there are some HM hyperaccumulators"   instead of  "In the other hand, there are some hyper HM accumulators"

Line 556: Correct: hyperaccumulate instead of hyper accumulate

Line 562:   Semicolon must be removed:   contents; accumulates

Line 588-591: Here is an obscure phrase. Is it possible to modify this sentence as follows? :

Moreover, in a crop rotation system, the use of a NFB-legume symbiosis product as a green manure before cultivation ofedible greens such as lettuce positively affected soil fertility due to a higher content of organic matter that quickly decomposed in the soil.

Line 621: Correct: "effective in improving"   instead of   "effective at improving"

Line 637-638: It is a noncompehensible sentence, check it out. Some words seem misused and a word is missing. The corrected the phrase is following (Does it still make sense?):MBPs increase accumulation and tolerance to HMs by PGPB through the synthesis of metal-binding proteins. 

Line 756: delete one comma: roots,,

Author Response

List of responses to each of the Reviewer #3 comments

We appreciate Reviewer’s suggestions. They are very valuable for our manuscript.

According to recommendations of Reviewer #3 we revised our manuscript to meet all requirements:

Line 16: Is one of the two words "directly" redundant? Rhizobia increase directly phytoremediation directly by nitrogen fixation ...

We removed one world “directly” from this sentence.

Line 32:  What did the authors mean by the following: wide-type or wild-type? ... both wide-type and genetically modified ...

We corrected the formula from “Wide type” to “Wild type” in the entire manuscript and corrected this part as follows: “…both wild type and genetically modified plants and bacteria…”

Line 112: Was "Be" missing in the next sentence:  legumes can [be] introduced

In line 137, we added the word “be” in the sentence.

Line 127-128: Was "comma or dot" missing at the end of the sentence?

The nature of pollution is quite varied

We added dot at the end of the sentence.

Line 213-214: Please check the meaning of the sentence. It does not seem to be created correctly.

Perhaps the authors meant the following:

Activation of the MT4 gene causes an increase in tolerance to metal ions, both Cu and Zn, in the leaves and roots of the host plant.

In line 239, we changed the sentence according to the reviewer’s recommendation.

Lines 269, 609, 633, 635, 637, 842: Did you meant "PGPR"? There is no "PGPB" in the list of abbreviations. 

It should be replaced by " PGPR" everywhere or "PGPB" should be added to abbreviations.

We added PGPB to abbreviation list and replaced by PGPB everywhere.

Line 396: The word "as" is missing: act [as] a symbiotic signal

In line 423, we added the word “as” in the sentence.

Line 401: Check for possible correction: "for acidic pH tolerance” instead of     "for tolerance of acidic pH"

In line 428, we corrected the formula according to reviewer’s recommendation.

Line 443: correct: "stimulating effect"   instead of    " stimuling effect"

In line 470, we corrected the formula according to reviewer’s recommendation.

Line 448: Is the word "to" missing: "related [to] the high production"

In line 475, we added the word “to” in the sentence.

Line 457: Correct: "species of Rhizobium increase"   instead of    "species of Rhizobium increases"

In line 482, we corrected the sentence according to reviewer’s recommendation.

Line 487:  Comma must be removed:  "total, leaf"

In line 522, we revised the sentence to make sense as the authors of the study intended as follows: The legume plant produced greater total plant, leaf, and root biomass.

Line 499-501:  The initial sentence is difficult to understant. It needs to be corrected, for example, as follows: "This procedure is likely to result in legumes and related rhizobia with sufficient inherent or selected resistance/tolerance to HMs to develop on phytostabilized soils contaminated with HMs."

Between lines 534-536, we corrected the sentence according to reviewer’s recommendation.

Line 549-550: Need to be corrected: "On the other hand, there are some HM hyperaccumulators"   instead of „In the other hand, there are some hyper HM accumulators"

Between lines 583-584, we corrected the sentence according to reviewer’s recommendation.

Line 556: Correct: hyperaccumulate instead of hyper accumulate

In line 590, we corrected the formula according to reviewer’s recommendation.

Line 562:   Semicolon must be removed:   contents; accumulates

In line 596, we removed the semicolon.

Line 588-591: Here is an obscure phrase. Is it possible to modify this sentence as follows?:

Moreover, in a crop rotation system, the use of a NFB-legume symbiosis product as a green manure before cultivation of edible greens such as lettuce positively affected soil fertility due to a higher content of organic matter that quickly decomposed in the soil.

Between 622 and 625, we corrected the sentence according to reviewer’s recommendation.

Line 621: Correct: "effective in improving"   instead of   "effective at improving"

In line 657, we corrected the formula according to reviewer’s recommendation.

Line 637-638: It is a noncompehensible sentence, check it out. Some words seem misused and a word is missing. The corrected the phrase is following (Does it still make sense?):MBPs increase accumulation and tolerance to HMs by PGPB through the synthesis of metal-binding proteins. 

Between lines 673-674, we corrected the sentence according to reviewer’s recommendation.

Line 756: delete one comma: roots,,

In line 795, we remove one comma after “root,”.

Moreover, the native speaker checked English language and style.